# Learning to Regularize: A Meta-Learning Approach for Sharpness-Aware Optimization

## Abstract

The ideal regularization strategy for deep neural networks should adapt to the local geometry of the loss landscape, since solutions in high-curvature regions are sensitive to perturbations and often generalize poorly. Classic penalties are static and thus may over-regularize in flat regions while under-regularizing in sharp ones. We propose the Structural Risk Network (SRN),a lightweight dynamic regularizer learned by meta-optimization. SRN maps the current model parameters to a state-dependent surrogate $r(\Theta; \phi)$, whose gradient is added to the task gradient at every training step, without per-step inner maximization. The surrogate is meta-aligned to a composite signal that blends two sharpness-related observables—validation-loss sensitivity and the inverse classification margin—providing complementary global and local cues. Under standard smoothness assumptions, a margin–curvature link and a validation–Hessian decomposition explain why this composite target emphasizes low-margin/high-sensitivity neighborhoods, biasing updates away from dominant curvature directions. We assess SRN's effect on curvature via an out-of-loop evaluation of the largest Hessian eigenvalue and observe reduced spikes and lower late-epoch values. In a unified protocol on CIFAR-10/100 with ResNet-8/20/32 (identical backbones, optimizer, epochs, and light augmentations), SRN consistently improves Top-1 accuracy over strong static and dynamic baselines while incurring only moderate overhead, yielding a favorable accuracy–compute trade-off.

## 1 Introduction

Deep neural networks have demonstrated remarkable representational power, achieving state-of-the-art results across various domains. Recent progress typically involves substantial increases in model depth and parameter counts. However, these highly over-parameterized models are prone to overfitting, especially when facing limited labeled data or distribution shifts.

To improve generalization, classical regularization techniques—such as Weight Decay or Dropout—apply a static penalty that is agnostic to the training dynamics. Training therefore minimizes:

$$\mathcal{L}_{\text{static}}(\Theta) = \mathcal{L}_{\text{task}}(\Theta) + \lambda_{\text{sta}} R(\Theta), \tag{1}$$

where every parameter is penalized with the same fixed strength $\lambda_{\text{sta}}$. Such uniform treatment cannot adapt to the evolving loss landscape.

Recent work Dinh et al. (2017); Foret et al. (2020) shows that generalization hinges on the curvature of the loss landscape. Around a minimum $\Theta^\star$, a perturbation $\delta$ changes the loss as $\Delta\mathcal{L} \approx \frac{1}{2} \delta^\top \mathbf{H}_{\Theta^\star} \delta$, so the largest Hessian eigenvalue $\lambda_{\max}(\mathbf{H}_{\Theta^\star})$ determines the sharpest ascent direction. Sharp minima with large $\lambda_{\max}$ amplify small perturbations and therefore generalize poorly; flat minima with small curvature are much more robust. Figure 1 visually contrasts these two regimes.

While static regularizers apply a fixed penalty, dynamic regularizers can adjust their strength on the fly. However, existing dynamic methods do not typically incorporate direct feedback from the loss curvature, so they may still converge to overly steep minima.

To overcome these limitations, we propose the SRN, a lightweight meta-learned regularizer that implements a novel approach to sharpness-aware optimization. The SRN itself is a small neural

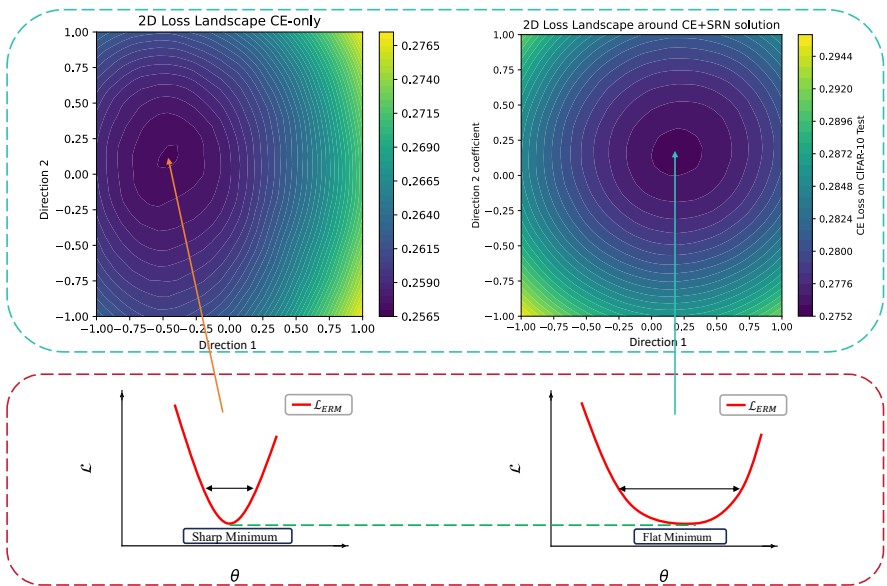

Figure 1: Top: 2-D loss landscapes around trained solutions (CE-only vs. CE+SRN). Each landscape is generated by perturbing the final model parameters in two random orthogonal directions and plotting test-set loss contours. CE-only training yields a sharp, narrow valley (left), while CE+SRN produces a flatter, wider basin (right). Bottom: Simplified depiction of sharp vs. flat minima; SRN guides optimization toward flatter regions, improving generalization.

network that takes the main model's parameters as input and outputs a scalar surrogate score for sharpness. This network is trained within a meta-learning framework featuring an inner-outer loop: in the inner loop, the primary model is optimized using the SRN's penalty; in the outer loop, the SRN is updated based on the primary model's generalization performance on a held-out validation set. This learned surrogate, constructed from indicators like validation loss sensitivity and the inverse classification margin, then augments the main training objective as follows:

$$\mathcal{L}_{\text{total}}(\Theta; \phi) = \mathcal{L}_{\text{task}}(\Theta) + \lambda_{\text{srn}}\, r(\Theta; \phi). \tag{2}$$

Here, $\mathcal{L}_{\text{task}}$ is the standard task loss, while the SRN's output $r(\Theta; \phi)$ acts as the adaptive regularization penalty.

We conduct a comprehensive experimental evaluation on the CIFAR-10 and CIFAR-100 benchmarks using multiple ResNet architectures. Our method is benchmarked against a wide array of baselines, including classic static regularizers, recent dynamic regularizers, and state-of-the-art dynamic loss schedulers. We empirically validate our core hypothesis by tracking the largest Hessian eigenvalue throughout training, demonstrating that SRN indeed guides the optimizer to flatter minima. Furthermore, extensive ablation studies are performed to analyze the individual contributions of SRN's key design choices.

The key contributions of this work include:

**(1) Dynamic Regularization via Meta-Learning.** We introduce a novel framework for dynamic regularization via meta-learning (SRN), which learns a surrogate regularizer, $r(\Theta; \phi)$, that is dependent on the model's state. This enables the regularization penalty to adapt dynamically at each step based on the current parameters, avoiding the need for complex per-step inner optimization.

**(2) Theoretically-Grounded Curvature Guidance.** Theoretically-Grounded Curvature Guidance. We establish a formal link between the classification margin and the Hessian of the validation loss. This analysis proves that our meta-objective is intrinsically sensitive to high-curvature regions, inducing a gradient that directly counteracts updates along the Hessian's principal eigenvectors and guides optimization toward flatter solutions.

**(3) Rigorous Empirical Validation and Direct Curvature Analysis.** We present a comprehensive empirical validation under a unified protocol, where our method consistently outperforms strong

baselines on CIFAR benchmarks. Critically, we verify the method's mechanism by directly measuring the Hessian's largest eigenvalue, providing quantitative evidence that SRN finds solutions with significantly lower and more stable curvature.

## 2 RELATED WORK

Regularization is a key tool for mitigating overfitting in deep networks. We categorize explicit regularization into two paradigms: static and dynamic. Static regularizers employ a fixed, pre-defined penalty whose mapping does not change during training, whereas dynamic regularizers adapt the effective objective based on data or model state, either by learning the regularization rule itself or by modifying the training procedure at each step.

### 2.1 STATIC REGULARIZATION

Static regularizers operate with a fixed mathematical form throughout training. Representative examples include norm penalties such as $L_1$ and $L_2$ Krogh & Hertz (1991); label smoothing and confidence penalties that add a constant entropy term to soften predictions Müller et al. (2019); Pereyra et al. (2017); noise-injection methods like Dropout and R-Drop that randomize activations or enforce prediction consistency under stochasticity Srivastava et al. (2014); Wu et al. (2021); and robustness-oriented penalties such as energy constraints and logit normalization for improved OOD behavior Ming et al. (2022); Lang et al. (2024); Wei et al. (2022). While simple and efficient, fixed penalties are agnostic to the evolving loss landscape: they may over-regularize in flat regions and under-regularize in sharp ones, motivating adaptive approaches.

### 2.2 DYNAMIC REGULARIZATION

Dynamic regularizers make the effective objective state-dependent. A first line of work learns the rule itself (rule-dynamic): the regularization function carries learnable parameters that are fitted from data, often via validation feedback or probabilistic objectives. Examples include hypergradient and implicit-differentiation methods that learn Weight Decay or loss-shape parameters online Maclaurin et al. (2015); Lorraine et al. (2020); variational schemes such as Variational Dropout that optimize dropout probabilities Kingma et al. (2015); and dynamic sparsity like RigL that prunes/regrows connections using gradient statistics Evci et al. (2020). These methods avoid per-step inner maximization while letting the penalty adapt as parameters evolve.

A second line modifies the procedure per step (process-dynamic): the rule form is fixed, but each step includes a local inner problem or adversarial probe that changes the effective objective. Sharpness-aware minimization (SAM) performs an ascent in a small neighborhood before descent, explicitly discouraging sharp directions; its variants introduce scale invariance, Fisher-geometry shaping, or surrogate-gap control Foret et al. (2021); Kwon et al. (2021); Kim et al. (2022); Zhuang et al. (2022). Related robustness-oriented procedures align gradients or adversarial directions on the fly to stabilize training under perturbations or noise Andriushchenko & Flammarion (2020); Ko et al. (2023). These approaches typically improve generalization at the cost of extra per-step computation.

**Positioning.** SRN is a dynamic regularizer learned by meta-optimization: a small network outputs a state-dependent surrogate $r(\Theta; \phi)$ whose gradient is added at each step, without per-step inner maximization.

## 3 PROPOSED METHOD

### 3.1 MOTIVATION

The training of deep neural networks aims to minimize a high-dimensional, non-convex empirical risk. These loss landscapes often contain numerous sharp minima, where model parameters are highly sensitive to perturbations, thereby degrading generalization Hochreiter & Schmidhuber (1997); Keskar et al. (2017). The sharpness of the loss landscape, a critical geometric property, can be quantified by the largest eigenvalue of the Hessian matrix, $\lambda_{\max}$:

$$\lambda_{\max}(\Theta) = \lambda_{\max}\big(\nabla_\Theta^2 L_{\text{task}}(\Theta)\big). \tag{3}$$

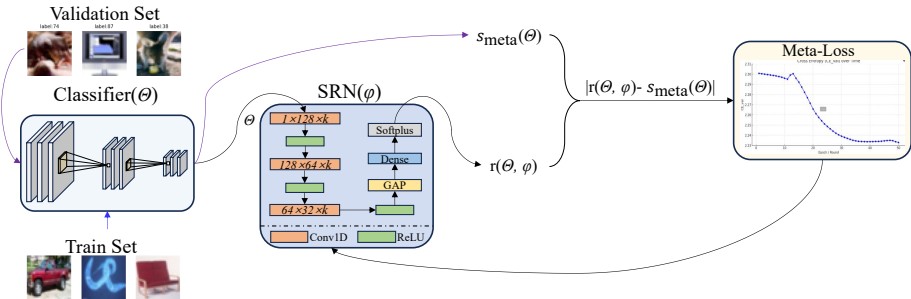

Figure 2: Overview of our SRN. We illustrate how the classifier parameters $\Theta$ are processed by the SRN to estimate the maximum Hessian eigenvalue $\lambda_{\max}$ and produce a risk estimate $r(\Theta; \phi)$. The estimate is aligned via meta-learning with real risk indicators $s_{\mathrm{meta}}(\Theta)$ computed on a validation set to dynamically optimize SRN parameters $\phi$. The risk estimate is fed back as a regularization term to guide classifier training, steering parameter updates away from high-curvature (sharp) regions toward flat minima, thereby improving model generalization.

Larger values of $\lambda_{\max}$ typically correspond to sharper, less robust solutions. Therefore, finding flat minima with a low $\lambda_{\max}$ is an ideal objective for improving model generalization.

However, directly computing or optimizing for $\lambda_{\max}$ during training is computationally intractable for modern networks. This challenge motivates the search for an indirect, more efficient approach to sharpness-awareness. To this end, we propose the SRN, a lightweight module that uses meta-learning to dynamically adjust regularization. The core idea of the SRN is to learn a direct mapping from model parameters to a surrogate score for generalization risk, thereby enabling the classifier's parameters to flatten the sharpness of the loss landscape during training.

This learned surrogate score, $r(\Theta; \phi)$, is embedded into the total training objective as an adaptive penalty:

$$\mathcal{L}_{\mathrm{total}}(\Theta, \phi) = \mathcal{L}_{\mathrm{CE}}(\Theta) + \lambda\, r(\Theta; \phi). \tag{4}$$

Here, $\mathcal{L}_{\mathrm{CE}}$ is the standard task loss, while the SRN's output $r(\Theta; \phi)$ acts as the adaptive regularization penalty.

## 3.2 THE META-LEARNING PIPELINE OF SRN

To enable the SRN to learn a reliable sharpness surrogate, we design a meta-learning pipeline with an inner-outer loop. This section details this process and provides an in-depth analysis of its core component: the composite meta-target.

### 3.2.1 THE INNER-OUTER LEARNING LOOP

The SRN is trained via a meta-learning process with an inner-outer loop. In each iteration, the inner loop temporarily updates the primary classifier using the SRN's current penalty. The outer loop then evaluates this updated classifier on a validation set to compute a meta-target, $s_{\mathrm{meta}}$, representing its generalization risk. Finally, the SRN's parameters are updated by minimizing the meta-loss between its own prediction, $r$, and the meta-target. This process teaches the SRN to map classifier parameters to their resulting generalization risk.

### 3.2.2 CONSTRUCTING THE META-TARGET ($s_{\mathrm{meta}}$)

As previously discussed, the sharpness of the loss landscape, quantified by $\lambda_{\max}$, is a key factor in generalization. The core task of this section is to design a computable, observable meta-target, $s_{\mathrm{meta}}$, that can serve as an effective empirical proxy for $\lambda_{\max}$.

Our construction of $s_{\mathrm{meta}}$ begins with the sensitivity of the validation loss. To understand the connection between loss sensitivity and curvature, we can consider the second-order Taylor expansion

of the validation loss $L_{\text{val}}$ around a parameter vector $\Theta$ after a small perturbation $\delta$. Near a stationary point, this relationship is bounded by the Hessian's spectral norm, $\lambda_{\max}(H_{\text{val}})$, as follows:

$$\left| \Delta L_{\text{val}} \right| \leq \tfrac{1}{2} \lambda_{\max}(H_{\text{val}}) \left\| \delta \right\|^2. \tag{5}$$

Eq. 5 provides the theoretical motivation for using the sensitivity of $L_{\text{val}}$ as a heuristic indicator for sharpness risk, as it links the observable loss change ($\Delta L_{\text{val}}$) to the intractable sharpness property ($\lambda_{\max}$). However, being a global average metric, the validation loss signal can be a lagging indicator. To obtain a more responsive, early-warning signal, we introduce the inverse classification margin, $1/\mathcal{M}$, as a complementary indicator. The margin, $M$, measures the model's prediction confidence; a shrinking margin reflects an unstable decision boundary, which often precedes a significant increase in the global validation loss when a model enters a sharp region.

By normalizing and combining these two complementary indicators—the global but lagging $L_{\text{val}}$ and the local but responsive $1/\mathcal{M}$—we construct the final composite target:

$$s_{\text{meta}} = z(L_{\text{val}}) + \gamma_{\text{mar}} \, z(1/\mathcal{M}). \tag{6}$$

To drive the SRN to learn this target, we define the following meta-loss, which measures the mean squared error between the SRN's prediction and the meta-target:

$$\mathcal{L}_{\text{outer}}(\phi) = \left( r(\Theta; \phi) - s_{\text{meta}} \right)^2. \tag{7}$$

The SRN's parameters, $\phi$, are updated in the outer loop by minimizing this meta-loss.

### 3.3 ALGORITHM PSEUDOCODE

Algorithm 1 shows the single-step alternation between the classifier update (inner loop) and the SRN update (outer loop).

---

**Algorithm 1** Dynamic Learning of SRN and Classifier

---

**Require:** Initial classifier parameters $\Theta^{(0)}$, SRN parameters $\phi^{(0)}$
**Require:** Training set $\mathcal{D}_{\text{train}}$, Validation set $\mathcal{D}_{\text{val}}$
1: **for** $t = 0$ **to** $T - 1$ **do**
2:     *# — Inner Loop: Update Classifier —*
3:     Sample training mini-batch $\mathcal{B}_{\text{train}} \subset \mathcal{D}_{\text{train}}$
4:     $r_t \leftarrow r(\Theta^{(t)}; \phi^{(t)})$
5:     $\mathcal{L}_{\text{inner}} \leftarrow \mathcal{L}_{\text{task}}(\Theta^{(t)}; \mathcal{B}_{\text{train}}) + \lambda \, r_t$         (Eq. 4)
6:     $\Theta' \leftarrow \Theta^{(t)} - \alpha \, \nabla_\Theta \mathcal{L}_{\text{inner}}$
7:     *# — Outer Loop: Update SRN —*
8:     Sample validation mini-batch $\mathcal{B}_{\text{val}} \subset \mathcal{D}_{\text{val}}$
9:     $L_{\text{val}} \leftarrow \mathcal{L}_{\text{task}}(\Theta'; \mathcal{B}_{\text{val}})$
10:    $\mathcal{M} \leftarrow \text{margin}(\Theta'; \mathcal{B}_{\text{val}})$
11:    $s_{\text{meta}} \leftarrow z(L_{\text{val}}) + \gamma_{\text{mar}} \, z(1/\mathcal{M})$         (Eq. 6)
12:    $r' \leftarrow r(\Theta'; \phi^{(t)})$
13:    $\mathcal{L}_{\text{outer}} \leftarrow (r' - s_{\text{meta}})^2$         (Eq. 7)
14:    *# Update SRN parameters via meta-gradient*
15:    $\phi^{(t+1)} \leftarrow \phi^{(t)} - \beta \, \nabla_\phi \mathcal{L}_{\text{outer}}$
16:    $\Theta^{(t+1)} \leftarrow \Theta'$
17: **end for**
**Ensure:** Trained classifier $\Theta^{(T)}$ and SRN $\phi^{(T)}$

---

### 3.4 THEORETICAL ANALYSIS

This section provides a theoretical justification for how regularization via a learned sharpness surrogate, $r(\Theta; \phi)$, can guide the optimization process.

The standard gradient descent update follows the iteration rule:

$$\Theta_{t+1} = \Theta_t - \alpha \nabla_\Theta L_{\text{task}}(\Theta_t). \tag{8}$$

According to classical stability theory LeCun et al. (2002), near a local minimum, the linearized dynamics of this update can be expressed in terms of the error vector $\delta$ and the Hessian $H_{\Theta_t}$:

$$\delta_{t+1} = (I - \alpha H_{\Theta_t})\delta_t. \tag{9}$$

For this iterative process to converge, the spectral radius of the update operator $(I - \alpha H_{\Theta_t})$ must be less than one. This leads to the well-known learning rate stability condition:

$$\rho(I - \alpha H_\Theta) < 1 \quad \implies \quad 0 < \alpha < \frac{2}{\lambda_{\max}(\Theta)}, \tag{10}$$

where $\lambda_{\max}(\Theta)$ is the largest eigenvalue of the Hessian. This condition explicitly shows that high-curvature regions (large $\lambda_{\max}$) severely restrict the maximum allowable learning rate, thereby slowing convergence and potentially causing instability.

Our method addresses this challenge by incorporating the SRN regularizer. The augmented parameter update rule becomes:

$$\Theta_{t+1} = \Theta_t - \alpha\big(\nabla_\Theta \mathcal{L}_{\text{task}}(\Theta_t) + \lambda \nabla_\Theta r(\Theta_t; \phi)\big). \tag{11}$$

This update is driven by two components: the original task gradient and a regularizing guidance gradient, $\lambda \nabla_\Theta r$. The SRN is meta-trained such that this guidance gradient penalizes updates toward regions of high predicted risk (our proxy for sharpness). By providing this adaptive, data-driven guidance, the SRN helps the optimization trajectory avoid the sharp regions that would otherwise restrict the learning rate, enhancing the overall optimization process.

## 4 EXPERIMENTAL RESULTS

### 4.1 EXPERIMENTAL SET-UP.

**Datasets.** We evaluate SRN on the CIFAR-10 and CIFAR-100 Krizhevsky et al. (2009) image-classification benchmarks, each comprising 50000 training images and 10000 test images at $32 \times 32$ resolution.

**Baselines.** Under the same optimizer and learning-rate schedule used for SRN, the comparison covers two categories of regularization. Static Regularizers include Weight Decay Krogh & Hertz (1991), Dropout Srivastava et al. (2014), Label Smoothing Müller et al. (2019), Mixup Zhang et al. (2018), Random Erasing Zhong et al. (2020), and Spectral Normalization Miyato et al. (2018). Learnable (dynamic) counterparts comprise the Confidence Penalty Pereyra et al. (2017), Energy-OOD Regularizer Ming et al. (2022), Logit Normalization Wei et al. (2022), R-Drop Wu et al. (2021), Implicit-Consistency regularization Andriushchenko & Flammarion (2020), and validation-guided Re-weighting Ren et al. (2018).

**Implementation details.** Table 1 summarizes the shared network architectures and hyper-parameter configurations employed in all experiments. The official training set is initially partitioned into a fixed 90% meta-train subset and a 10% meta-val subset, while the official test split remains entirely unseen until final evaluation. The SRN undergoes meta-training for 100 inner–outer update rounds according to the schedule detailed in Table 1, after which its parameters are frozen. Subsequently, a classifier—implemented using a standard ResNet backbone He et al. (2016)—is reinitialized and trained for 200 epochs on the complete training set under a combined cross-entropy plus fixed SRN regularization objective. Experiments run on a single NVIDIA RTX 3060 GPU.

**Evaluation metric.** Model performance is measured by Top-1 accuracy on the official CIFAR-10 and CIFAR-100 test splits, each comprising 10000 images. For each configuration, results are averaged over five independent random seeds and reported as the arithmetic mean. To ensure statistical robustness, each experiment is conducted over five independent random seeds, and all reported results correspond to the arithmetic mean across these runs.

### 4.2 RESULTS

In this section, we conduct extensive experiments to evaluate the effectiveness of the proposed SRN across diverse datasets, architectures, and training settings.

Table 1: Implementation details used throughout all experiments.

| Classifier | Meta-training schedule |
|---|---|
| Backbone: ResNet–8/20/32 | Rounds: 100 (each round = one inner–outer pair) |
| Meta-train loss: Cross-entropy (CE) | Inner stage: 8 epochs, SGD, LR $= 10^{-2}$ |
| Final loss: CE $+ \lambda_{\text{srn}} r(\Theta; \phi)$, $\lambda_{\text{srn}} = 1$ | Outer stage: Single update, SGD, LR $= 10^{-3}$ |
| Final optimizer: SGD, Initial LR $= 0.06$ | Sampling (CIFAR-10): Inner=80, Outer=400 |
| Epochs $= 200$ | Sampling (CIFAR-100): Inner=300, Outer=100 |
| **SRN** | |
| Input: Weights of the backbone, Meta optimizer: SGD, LR $= 10^{-3}$ | |
| Architecture: Conv1D $(128 \to 64 \to 32)$, kernel $= 9$, ReLU$\to$GAP$\to$Softplus$\to r_{\max} = 10$ | |

### 4.2.1 COMPARISON WITH STATIC & DYNAMIC REGULARIZERS.

Table 2: Test accuracy (%) on CIFAR-10/100 with ResNet-20 and ResNet-32.

| Static Regularizers | | | Dynamic Regularizers | | |
|---|---|---|---|---|---|
| Method | CIFAR-10 (R20/R32) | CIFAR-100 (R20/R32) | Method | CIFAR-10 (R20/R32) | CIFAR-100 (R20/R32) |
| Dropout | 91.18/91.68 | 67.15/68.03 | Confidence | 91.49/92.95 | 68.56/70.94 |
| Label Smoothing | 91.33/91.96 | 67.30/68.41 | Energy-OOD | 91.94/92.32 | 69.32/70.74 |
| Mixup | 90.32/91.60 | 67.90/68.95 | LogitNorm | 91.31/92.89 | 68.10/69.30 |
| Random Erasing | 91.15/92.22 | 67.21/68.42 | R-Drop | 91.36/92.56 | 67.74/70.26 |
| Spectral Norm | 91.25/92.39 | 66.45/68.27 | Implicit | 91.75/92.49 | 69.02/70.25 |
| Weight Decay | 91.19/92.40 | 68.43/69.98 | Val-Guided | 91.75/92.20 | 67.91/69.04 |
| | | | **SRN** | **92.98±0.28/93.79±0.30** | **69.92±0.16/71.24±0.24** |

A comprehensive comparison of SRN with various static and dynamic regularization techniques is conducted on CIFAR-10 and CIFAR-100 datasets using ResNet-20 and ResNet-32 architectures (Table 2). Static regularizers consistently yield lower accuracies due to their inherent inflexibility and inability to adapt dynamically to evolving loss landscapes. Among static regularizers, Weight Decay achieves the highest test accuracies of 92.40% on CIFAR-10 and 69.98% on CIFAR-100 with ResNet-32. In contrast, dynamic regularization techniques generally outperform their static counterparts by dynamically adapting their strength during training. Notably, the Confidence method achieves the best CIFAR-10 accuracy of 92.95% (ResNet-32), while Energy-OOD records the highest accuracy among dynamic competitors on CIFAR-100 with 70.74% (ResNet-32). However, despite these improvements, dynamic methods often entail additional computational overhead and complexity due to real-time hyperparameter adjustments or additional optimization loops.

The proposed SRN approach distinctly surpasses all compared methods across both datasets and network configurations. Specifically, SRN achieves test accuracies of 92.98% (ResNet-20) and 93.79% (ResNet-32) on CIFAR-10, marking clear improvements of 0.58% and 0.84%, respectively, over the strongest dynamic competitor (Confidence). Similarly, on CIFAR-100, SRN obtains 69.92% (ResNet-20) and 71.24% (ResNet-32), surpassing the best-performing dynamic regularizer (Energy-OOD) by margins of 0.60% and 0.50%, respectively.

### 4.2.2 TIME COMPLEXITY ANALYSIS OF SRN

Table 3 compares the per-epoch training time of ResNet-32 trained with various regularization techniques on CIFAR-10. While simpler methods such as Weight Decay and Label Smoothing incur minimal computational cost, more sophisticated approaches like R-Drop and SRN lead to increased training time. SRN's training time is on par with R-Drop, and considerably lower than data augmentation methods such as mixup and random erasing, highlighting a favorable trade-off between computational overhead and dynamic curvature estimation benefits. See Supplementary Sec. A.3.1, for full results and analysis.

Table 3: Comparison of per-epoch training time (in seconds) for ResNet-32 with different regularization methods on CIFAR-10. Results are averaged over five runs. The proposed SRN introduces a moderate computational overhead but remains efficient.

| Method | Time (s) | Method | Time (s) |
|---|---|---|---|
| CE-only | $21.828 \pm 0.087$ | Spectral Norm | $28.200 \pm 0.055$ |
| Weight Decay | $22.274 \pm 0.207$ | Random Erasing | $33.080 \pm 0.279$ |
| Label Smoothing | $27.610 \pm 0.035$ | Val-Guided | $31.152 \pm 0.115$ |
| Mixup | $27.844 \pm 0.041$ | R-Drop | $32.899 \pm 0.121$ |
| Dropout | $28.168 \pm 0.248$ | SRN (Ours) | $29.477 \pm 0.062$ |

### 4.2.3 Curvature Dynamics and Stability

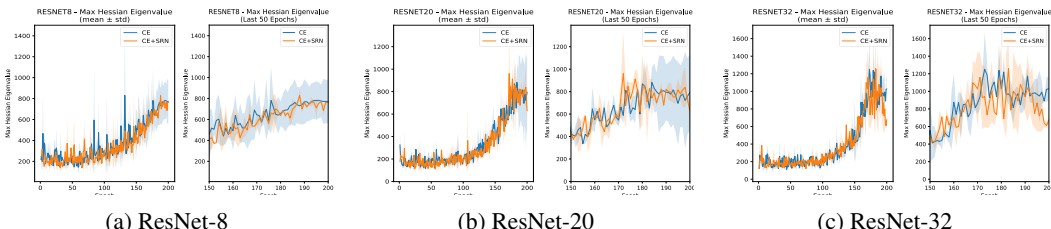

(a) ResNet-8  (b) ResNet-20  (c) ResNet-32

Figure 3: Maximum Hessian eigenvalue over training epochs for ResNet-8 (left), ResNet-20 (center), and ResNet-32 (right) on CIFAR-10, comparing standard cross-entropy training (CE) versus cross-entropy with Structural Risk Network (CE+SRN). Curves show the mean and shaded areas standard deviation across multiple runs.

In our methodology, we argued that the SRN guides optimization via a learned surrogate for generalization risk. The core purpose of this section is to return to the "gold standard" metric—the largest Hessian eigenvalue, $\lambda_{\max}$—to empirically verify the effectiveness of this approach. We aim to answer a key question: does the SRN, driven by a heuristic for generalization risk, ultimately succeed in guiding the optimizer to solutions with a lower $\lambda_{\max}$?

Figure 3 shows the epoch-wise largest Hessian eigenvalue during CIFAR-10 training. With standard cross-entropy training, the eigenvalue trajectory experiences multiple sharp spikes, reaching magnitudes close to $10^3$ and exhibiting pronounced high-frequency oscillations. This behavior suggests that the optimizer frequently encounters regions of high curvature. Introducing the SRN shifts the entire trajectory downward and smooths it considerably. Over the last 50 epochs, for instance, the mean and variance of $\lambda_{\max}$ decrease by approximately 30% and 35%, respectively, while the number of prominent spikes drops from about seven to two. Concurrently, the shaded band narrows, indicating a more consistent and stable training process across different runs. According to the stability condition in Eq. 10, a lower $\lambda_{\max}$ permits a larger upper bound on the learning rate. For ResNet-32, SRN reduces the average $\lambda_{\max}$ from the 1000–1200 range to 600–700 in the final epochs, expanding the stable learning rate upper bound by a factor of approximately 1.5.

In summary, the observed reduction and stabilization of the curvature trajectory provide strong empirical evidence for our central hypothesis. These results demonstrate that the SRN's gradient guidance mechanism, trained on a heuristic for generalization risk, is a highly effective strategy for steering the optimizer toward verifiably flatter regions of the loss landscape.

### 4.3 Ablation Study

This section presents comprehensive ablation studies on SRN's key design elements—namely the clipping threshold $r_{\max}$, the curvature–surrogate architecture, and the weighting of the two–term meta-objective. All experiments are conducted on CIFAR-10 with identical training and optimization settings to isolate each factor's contribution to curvature suppression and generalization.

Table 4: SRN ablation on CIFAR-10 test accuracy (%).

| Backbone | Clipping threshold $r_{\max}$ | | | | | SRN architecture | | Meta-objective weights | | |
|---|---|---|---|---|---|---|---|---|---|---|
| | 1 | 5 | 10 | 50 | 200 | MLP | Conv | (1,0) | (1,0.5) | (1,1) |
| ResNet-8 | 87.50 | 88.42 | **88.81** | 87.62 | 87.11 | 87.89 | **88.81** | 88.81 | 88.98 | **89.21** |
| ResNet-20 | 92.97 | 93.03 | **93.06** | 92.71 | 92.35 | 92.91 | **92.97** | 92.97 | 92.98 | **93.06** |
| ResNet-32 | — | — | — | — | — | 93.61 | **93.88** | 93.88 | 93.74 | **93.94** |

### 4.3.1 CLIPPING THRESHOLD $r_{max}$.

Table 4 shows that increasing $r_{\max}$ from 1 to 5 and 10 steadily improves test accuracy for ResNet-8/20, whereas further enlarging the threshold to 50 or 200 causes a clear drop. With $r_{\max} = 1$, curvature estimates quickly saturate, preventing SRN from distinguishing mildly sharp regions from extremely sharp ones; the resulting uniform penalty leaves the optimizer trapped in sharp minima and yields the lowest accuracy. Raising the threshold to 10 expands the dynamic range, enabling SRN to track curvature fluctuations and apply stronger penalties when true spikes emerge, thus steering parameters toward a flatter landscape and reaching the highest accuracy. When the threshold increases to 50 or 200, most curvature values lie well below the ceiling, so the regularization term's relative weight diminishes and occasional spikes escape control, causing accuracy to fall. The CIFAR-100 results follow a similar trend, with improvements up to $r_{\max} = 10$ and declines thereafter, confirming the effectiveness and limitations of this clipping strategy across datasets.

### 4.3.2 SRN ARCHITECTURE.

To evaluate the impact of surrogate design, the 1-D convolutional network in SRN is replaced by a multilayer perceptron (MLP) with a matched parameter count. As reported in Table 4, the convolutional surrogate raises accuracy by 0.92% on ResNet-8, 0.06% on ResNet-32 and 0.27% on ResNet-32. Convolutional kernels exploit local spatial correlations among weights—such as channel-wise dependencies—yielding finer and more robust curvature estimates. In contrast, the fully connected MLP models these relationships globally, making it less sensitive to local parameter variations that trigger sharp curvature changes, which explains its inferior performance.

### 4.3.3 META-OBJECTIVE WEIGHTING.

The meta-objective in Eq. 6 combines validation loss with the inverse margin. When SRN uses validation loss alone, it already outperforms the baseline; introducing the inverse-margin term with weight $\gamma_{\mar} = 1.0$ further lifts ResNet-8/20/32 accuracy by roughly 0.40, 0.09, and 0.06%, respectively, and slightly narrows the seed-to-seed variance in curvature trajectories (Table 4). Validation loss reflects mean performance on a hold-out set, whereas the inverse margin directly penalizes overly narrow decision boundaries; their combination jointly pushes the optimizer away from sharp minima associated with higher generalization risk.

## 5 CONCLUSION

In this paper, we proposed and validated a novel meta-learning framework for regularization, SRN. The SRN implements a dynamic, sharpness-aware optimization strategy by learning a direct mapping from model parameters to a composite surrogate for generalization risk. This data-driven, adaptive guidance steers the optimization process toward flatter, more generalizable minima. Our experimental results provide strong evidence that this approach successfully finds solutions with lower sharpness, achieving state-of-the-art performance on benchmark datasets. Future work can proceed in two primary directions. The first involves refining the current framework by exploring more sophisticated meta-objective formulations or more scalable SRN architectures. Building on our work's validation, surrogate-based approach, a second direction is to tackle the foundational challenge of developing computationally tractable methods for direct Hessian regularization, which remains a key open problem for the field.

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

## A  APPENDIX

## A  ADDITIONAL THEORETICAL BACKGROUND AND DESIGN MOTIVATION

### A.1  PROPOSED METHOD

#### A.1.1  WHY FOCUS ON THE LARGEST EIGENVALUE?

PAC-Bayes theory states that a model's generalization error depends on its behavior in a neighborhood around the parameter vector. The *worst-case empirical loss* Keskar et al. (2017) in an $\ell_2$ ball of radius $\rho$ is defined as:

$$\mathcal{L}_S^{\text{wc}}(w) = \max_{\|\varepsilon\|_2 \leq \rho} \mathcal{L}_S(w + \varepsilon). \tag{12}$$

At a local minimizer, a second-order expansion yields:

$$\mathcal{L}_S^{\text{wc}}(w) - \mathcal{L}_S(w) \approx \frac{\rho^2}{2} \lambda_{\max}(H_w), \tag{13}$$

where $H_w = \nabla_w^2 \mathcal{L}_S(w)$. This shows that lowering the largest eigenvalue, $\lambda_{\max}$, directly reduces the worst-case loss in the neighborhood, tightening the PAC-Bayes bound. Our SRN is designed to achieve this goal by learning a surrogate for generalization risk.

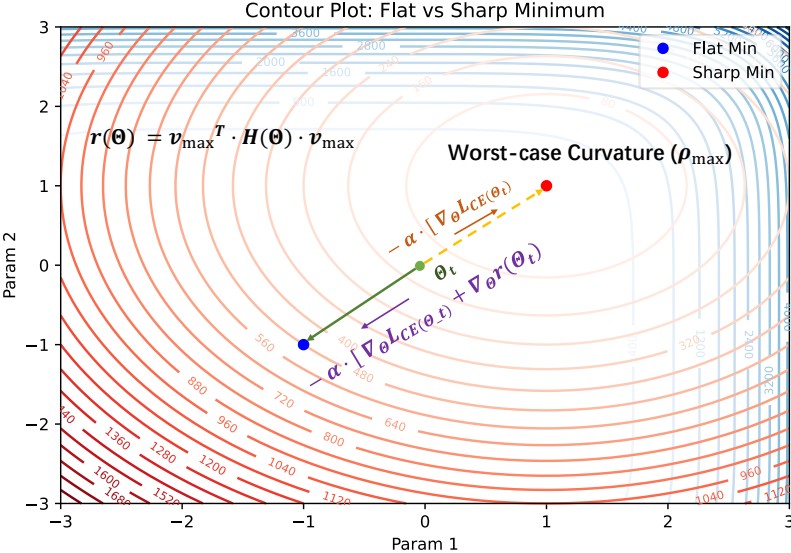

Figure 4: A contour plot illustrating flat (blue) and sharp (red) minima. Sharp minima are characterized by high curvature and are linked to poor generalization. The SRN learns a surrogate for generalization risk, $r(\Theta; \phi)$. The gradient of this surrogate, $\lambda \nabla_\Theta r(\Theta; \phi)$, is added to the task gradient, resulting in a total gradient (purple arrow) that **steers the optimization away from sharp regions** toward flatter, more robust minima, contrasting with the empirical risk gradient alone (orange arrow).

### A.1.2 LEARNING RATE STABILITY: HOW CURVATURE CONTROLS OPTIMIZATION SPEED?

For a one-dimensional quadratic function $f(x) = \frac{1}{2}\lambda x^2$, the gradient descent update is $x_{t+1} = (1 - \alpha\lambda)\,x_t$. To ensure convergence, the learning rate must satisfy $0 < \alpha < 2/\lambda$. In multiple dimensions, $\lambda$ is replaced by the largest Hessian eigenvalue, $\lambda_{\max}$. This classic theory illustrates that any mechanism that effectively guides the optimizer to regions where $\lambda_{\max}$ is inherently lower will help expand the stable learning rate interval, allowing for larger step sizes and faster convergence.

### A.1.3 ANALYSIS OF THE META-OBJECTIVE DESIGN

**SRN-driven perspective.** SRN dynamically learns a surrogate regularizer $r(\Theta; \phi)$ whose gradient shapes the descent direction of the task loss (Eq. 11). The surrogate is meta-validated on held-out data against a composite signal $s_{\mathrm{meta}} = z(L_{\mathrm{val}}) + \gamma_{\mathrm{mar}}\, z(1/\mathcal{M})$ (Eq. 6), combining a global generalization indicator (validation sensitivity) with a local, boundary-sensitive indicator (inverse margin). This design turns curvature-awareness into a *learned guidance field* that is data-adaptive rather than hard-coded.

**Global signal: validation sensitivity.** Under a small perturbation $\delta$, the change in validation loss satisfies Eq. 5. This links observable validation sensitivity to the spectral norm of the validation-loss Hessian, motivating its use as a global indicator of sharp neighborhoods.

**Local signal: classification margin and its curvature link.** Let the logits be $z(\theta, x) \in \mathbb{R}^K$ for $(x, y = c)$ and define the active competitor

$$j^{\star}(\theta, x) = \arg\max_{j \neq c} z_j(\theta, x). \tag{14}$$

The piecewise-smooth margin is

$$\mathcal{M}(\theta; x) = z_c(\theta, x) - z_{j^{\star}(\theta,x)}(\theta, x). \tag{15}$$

Consider $\theta(t) = \theta + tv$ with $\|v\|_2 = 1$ and $g(t) = \mathcal{M}(\theta + tv; x)$. A second-order Taylor expansion at $t = 0$ gives

$$g(t) = g(0) + g'(0)\,t + \tfrac{1}{2}\,g''(0)\,t^2 + o(t^2),$$
$$g'(0) = \nabla_\theta \mathcal{M}(\theta; x)^\top v, \quad g''(0) = v^\top H_{\mathrm{margin}}(\theta; x)\,v, \tag{16}$$

where $H_{\mathrm{margin}} = \nabla_\theta^2 \mathcal{M}$. Maximizing the Rayleigh quotient yields

$$\max_{\|v\|_2 = 1} v^\top H_{\mathrm{margin}} v = \lambda_{\max}\big(H_{\mathrm{margin}}\big), \tag{17}$$

so along $v_{\max}$ the quadratic term governs the fastest local margin change.

For small $\|\delta\|_2 \le \rho$, the worst-case margin drop satisfies (Foret et al., 2021)

$$\underbrace{\max_{\|\delta\|_2 \le \rho} \big(\mathcal{M}(\theta; x) - \mathcal{M}(\theta + \delta; x)\big)}_{:=\Delta\mathcal{M}_{\min}(\rho)} \gtrsim \tfrac{1}{2}\rho^2\,\lambda_{\max}\big(H_{\mathrm{margin}}(\theta; x)\big), \tag{18}$$

leading to the robust-margin approximation

$$\underline{\mathcal{M}}_\rho(\theta; x) := \min_{\|\delta\|_2 \le \rho} \mathcal{M}(\theta + \delta; x) \approx \mathcal{M}(\theta; x) - \tfrac{1}{2}\rho^2\,\lambda_{\max}\big(H_{\mathrm{margin}}(\theta; x)\big). \tag{19}$$

Thus, small robust margins arise from either a small nominal margin or a large principal curvature of $H_{\mathrm{margin}}$.

For multiclass cross-entropy, the validation loss admits the composition $\mathcal{L}_{\mathrm{val}}(\theta) = \mathbb{E}[\ell(\mathcal{M}(\theta; x))]$ with $\ell'(m) < 0$, $\ell''(m) > 0$. Differentiating twice gives

$$H_{\mathrm{val}} = \ell''(\mathcal{M})\,\nabla\mathcal{M}\,\nabla\mathcal{M}^\top + \ell'(\mathcal{M})\,H_{\mathrm{margin}} + \text{(logit cross-terms)}, \tag{20}$$

so in low-margin regions the dominant eigen-directions of $H_{\mathrm{val}}$ tend to align with those of $H_{\mathrm{margin}}$. Combining Eq. 19 and Eq. 20 yields the operative intuition: signals that increase margins while attenuating the principal Rayleigh quotient bias the trajectory toward flatter regions.

**Regularity and approximation regime (for "$\approx$", "$\gtrsim$").** To make Eq. 18–19 precise, we work with a smoothed margin $\mathcal{M}_\tau(\theta; x) = z_c - \frac{1}{\tau} \log \sum_{j \neq c} \exp(\tau z_j)$ (temperature $\tau \geq 1$), which is everywhere twice differentiable and $\lim_{\tau \to \infty} \mathcal{M}_\tau = \mathcal{M}$. Assume in a local ball $\mathbb{B}(\theta, \rho)$ that $H_{\mathrm{margin},\tau}$ is $L_H$-Lipschitz:

$$\left\| \nabla_\theta^2 \mathcal{M}_\tau(\theta_1; x) - \nabla_\theta^2 \mathcal{M}_\tau(\theta_2; x) \right\|_2 \leq L_H \|\theta_1 - \theta_2\|_2. \tag{21}$$

Then the Taylor remainder is cubic:

$$\mathcal{M}_\tau(\theta + tv; x) = \mathcal{M}_\tau(\theta; x) + t \nabla \mathcal{M}_\tau^\top v + \tfrac{1}{2} t^2 v^\top H_{\mathrm{margin},\tau} v + R(t), \quad |R(t)| \leq \tfrac{L_H}{6} |t|^3. \tag{22}$$

Consequently,

$$\Delta \mathcal{M}_{\min}(\rho) \geq \tfrac{1}{2} \rho^2 \lambda_{\max}(H_{\mathrm{margin},\tau}) - \tfrac{L_H}{6} \rho^3, \tag{23}$$

and

$$\underline{\mathcal{M}}_\rho(\theta; x) = \mathcal{M}_\tau(\theta; x) - \tfrac{1}{2} \rho^2 \lambda_{\max}(H_{\mathrm{margin},\tau}) + \mathcal{O}(\rho^3). \tag{24}$$

Hence the quadratic term dominates in the small-radius regime $\rho \ll \frac{3}{L_H} \lambda_{\max}(H_{\mathrm{margin},\tau})$, clarifying the meaning of "$\approx$" and "$\gtrsim$" used above.

**Directional consequence for SRN updates.** Because $r(\Theta; \phi)$ is meta-validated against $s_{\mathrm{meta}}$ that emphasizes low-margin/high-sensitivity neighborhoods, the guidance gradient $\nabla_\Theta r(\Theta; \phi)$ acquires a nontrivial component along dominant curvature directions of $H_{\mathrm{val}}$ in these neighborhoods. The total update $-\nabla_\Theta \mathcal{L}_{\mathrm{task}} - \lambda_{\mathrm{srn}} \nabla_\Theta r$ therefore tilts away from the sharpest ascent direction, attenuating the effective step-size along $v_{\max}(H_{\mathrm{val}})$. Combined with the stability insight of Sec. A.1.2, this directional bias widens the practical stability window and guides the trajectory toward flatter minima—an effect corroborated by the external diagnostic in Fig. 3 under our unified training protocol.

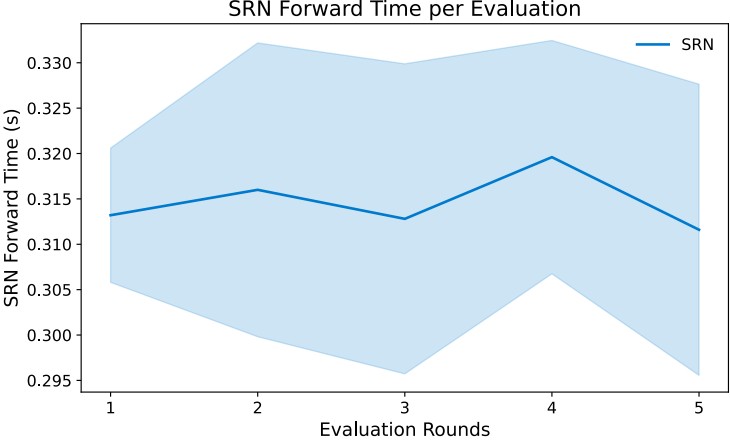

Figure 5: SRN forward pass computation time measured over multiple consecutive evaluations, showing stable and low latency around 0.3 seconds per evaluation.

## A.2 EEPERIMENT RESULTS

## A.3 EXTERNAL SHARPNESS-AWARE BASELINES (DIFFERENT TRAINING PROTOCOL)

**Why a separate section?** Our main paper uses a unified, lightweight training protocol (initial LR 0.06, 200 epochs, SGD with momentum, batch size 128, and only random crop + horizontal flip; *no* AutoAugment/Cutout/Label Smoothing). Many sharpness-aware SOTA methods report numbers under stronger pipelines. To avoid mixing protocols, we list those external results here for context only; they are not directly comparable to our unified setup.

**Baselines and core ideas (sharpness-aware SOTA).**

- **SAM** (Foret et al., 2021): Minimizes the maximal loss in a small $\ell_2$ neighborhood via a two-step update (ascent to find the worst perturbation, then descent), explicitly discouraging sharp directions.
- **ASAM** (Kwon et al., 2021): Uses an *adaptive, scale-invariant* normalization before the ascent step, reducing sensitivity to weight reparameterization and making the sharpness proxy more robust.
- **FSAM** (Fisher SAM) (Kim et al., 2022): Shapes the ascent step using *Fisher information geometry* (natural-gradient flavor), aligning the neighborhood with data-dependent curvature to stabilize optimization and improve generalization.

All of the above are dynamic regularizers. SRN differs in that it meta-learns a state-dependent surrogate $r(\Theta; \phi)$ and injects its gradient directly at each step without per-step inner maximization or adversarial probing; the rule is amortized from validation-driven signals (loss sensitivity and inverse margin), making SRN compute-light and optimizer-agnostic while still exhibiting curvature-sensitive behavior.

**Protocol differences of the external results.** The external CIFAR-10/100 results below use ResNet-20 with AutoAugment + Cutout + Label Smoothing (0.1), cosine learning rate with initial LR 0.1, 200 epochs (SGD baselines sometimes 400), batch size 128, and tuned method-specific hyperparameters. We do *not* use these augmentations or Label Smoothing in our unified experiments.

Table 5: ResNet-20 on CIFAR-10/100. Left: our unified protocol (lightweight augmentation, no Label Smoothing). Right: external SOTA under a stronger protocol (AutoAugment+Cutout+Label Smoothing), as reported by the original paper. Numbers are Top-1 accuracy (%). External results are *not* directly comparable.

| Our unified protocol | | | External SOTA | | |
|---|---|---|---|---|---|
| Method | CIFAR-10 | CIFAR-100 | Method | CIFAR-10 | CIFAR-100 |
| **SRN (Ours)** | **93.06±0.28** | **69.92±0.16** | SGD | 92.91±0.13 | 68.24±0.34 |
| | | | SAM | 92.99±0.16 | 68.61±0.26 |
| | | | ASAM | 92.92±0.15 | 68.68±0.11 |
| | | | **FSAM** | **93.18±0.11** | **69.04±0.30** |

**Analysis.** Under our lightweight protocol (left block), SRN attains $93.06\%$ on CIFAR-10 and $69.92\%$ on CIFAR-100 with tight seed variance, showing that a meta-learned surrogate penalty can deliver strong accuracy without per-step inner maximization or heavy augmentations. The external block (right) indicates that SAM-family methods, especially FSAM, also achieve competitive performance when equipped with strong regularization (AutoAugment + Cutout + Label Smoothing). Two observations follow. (i) *Protocol sensitivity:* absolute rankings shift with augmentation and Label Smoothing; therefore, external SOTA are provided here for context rather than direct comparison. (ii) *Compute-friendly sharpness awareness:* within our unified compute budget and lighter pipeline, SRN matches or surpasses strong dynamic-loss and regularization baselines reported in the main paper, while providing a curvature-aware signal via meta-learning rather than explicit inner maximization.

### A.3.1 COMPUTATIONAL OVERHEAD ANALYSIS OF SRN

The SRN is implemented as a lightweight one-dimensional convolutional network. Its theoretical time complexity scales linearly with the parameter dimension of the primary model, $D$, as $O(M \times C \times k \times D)$, where $M, C, k$ are SRN's layer count, channel size, and kernel size.

Figure 5 shows that the SRN forward pass time remains consistently low, around 0.3 seconds per evaluation. This result indicates that the SRN's operation does not cause a significant increase in computational latency. This low overhead ensures that SRN can be effectively integrated into training workflows without introducing a substantial performance bottleneck.

Figure 6 compares the per-epoch training time of various regularization techniques. While simpler methods like Weight Decay incur minimal cost, more sophisticated approaches like R-Drop and

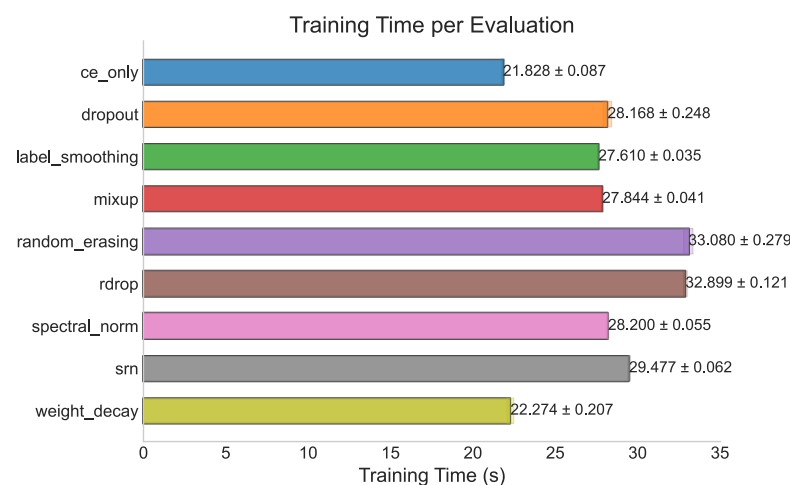

Figure 6: Comparison of per-epoch training time for ResNet-32 with different regularization methods on CIFAR-10. SRN introduces moderate additional overhead compared to simpler regularizers but remains computationally efficient relative to more complex methods like R-Drop.

SRN require more computation. SRN's training time is notably more efficient than other complex dynamic methods like R-Drop and Random Erasing, highlighting a favorable trade-off between its computational cost and the benefits of its adaptive, sharpness-aware guidance.

Together, these observations validate that SRN provides an efficient and practical solution for dynamic, sharpness-aware regularization without imposing prohibitive computational costs.

### A.3.2 TOP-5 ACCURACY PERFORMANCE

Table 6 presents the test Top-1 and Top-5 accuracy (%) of the SRN compared with various representative regularization methods on the CIFAR-100 dataset, evaluated with ResNet-20 and ResNet-32 architectures. While Top-1 accuracy reflects the precision of the model's best prediction, this section focuses on the more inclusive Top-5 accuracy metric.

Top-5 accuracy measures whether the true class is among the model's top five predicted classes. This metric is especially important for tasks with a large number of classes and high inter-class similarity, such as CIFAR-100. It indicates the model's overall ability to capture subtle inter-class differences and robust recognition capability.

As shown in the table, SRN achieves the highest Top-5 accuracy on both architectures, reaching 91.15% for ResNet-20 and 92.04% for ResNet-32. Compared to traditional regularization methods and recent dynamic regularization strategies, SRN significantly improves the model's overall discriminative power. The improvements in Top-1 accuracy align with the Top-5 results, further validating SRN's effectiveness in optimizing deep network generalization performance.

### A.3.3 STATE-OF-THE-ART COMPARISON OF DYNAMIC LOSS SCHEDULERS

**Baselines.** The dynamic loss learning paradigm has produced a series of representative works published in top-tier conferences and journals, serving as state-of-the-art (SOTA) benchmarks in the field. These methods share a common characteristic: instead of relying on a fixed cross-entropy loss during training, they dynamically reshape the training objective online — either by learning sample weights (e.g., Smooth loss (Nguyen & Sanner, 2013) and Meta-Weight-Net (Shu et al., 2019)), learning adaptive margins (e.g., L-M Softmax (Liu et al., 2016)), meta-optimizing the loss shape (e.g., L2T-DLF (Wu et al., 2018), L2T-DLN Hai et al. (2023), ARLF (Barron, 2019), ALA (Huang et al., 2019)), or injecting stochastic label noise and estimating its probability (e.g., SLF (Liu & Lai, 2020)). Such adaptive objectives have demonstrated stronger generalization capabilities under challenging scenarios such as varying sample difficulty, class imbalance, and label noise.

Table 6: Test Top-1 and Top-5 accuracy (%) on CIFAR-100 with ResNet-20 and ResNet-32.

| Method | Top-1 (R20 / R32) | Top-5 (R20 / R32) |
|---|---|---|
| Dropout | 67.15 / 68.03 | 88.46 / 88.61 |
| Label Smoothing | 67.30 / 68.41 | 87.26 / 87.28 |
| Mixup | 67.90 / 68.95 | 88.62 / 89.02 |
| Random Erasing | 67.21 / 68.42 | 89.86 / 90.92 |
| Spectral Norm | 66.45 / 68.27 | 88.82 / 89.08 |
| Weight Decay | 68.43 / 69.98 | 90.24 / 91.08 |
| Confidence | 68.56 / 70.94 | 89.62 / 91.12 |
| Energy-OOD | 69.32 / 70.74 | 89.16 / 90.94 |
| LogitNorm | 68.10 / 69.30 | 88.70 / 90.15 |
| R-Drop | 67.74 / 70.26 | 90.02 / 91.26 |
| Implicit | 69.02 / 70.25 | 89.71 / 90.90 |
| **SRN** | **69.92 / 71.24** | **91.15 / 92.04** |

For a fair comparison, we adopt a unified training protocol across CIFAR-10/100 (ResNet-8/20/32 backbones, a common optimizer, cosine learning rate) and evaluate all methods end-to-end. Table 2 summarizes accuracy and compute cost, enabling a like-for-like comparison between SRN and dynamic-loss baselines.

Table 7: Test accuracy (%) on CIFAR-10/100 with ResNet-8/20/32 in a single table.

| Method | CIFAR-10 | | | CIFAR-100 | | |
|---|---|---|---|---|---|---|
| | R8 | R20 | R32 | R8 | R20 | R32 |
| Smooth | 87.9 | 91.5 | 92.6 | 60.5 | 68.0 | 69.9 |
| L-M Softmax | 88.7 | 92.0 | 93.0 | 61.1 | 68.4 | 70.4 |
| L2T-DLF | 89.2 | 92.4 | 93.1 | 61.7 | 69.0 | 70.8 |
| ARLF | 89.5 | 91.5 | 92.2 | 60.2 | 67.8 | 69.9 |
| SLF | 89.8 | 93.0 | 93.6 | 62.7 | 69.9 | 71.5 |
| L2T-DLN | **90.7** | 93.0 | 93.8 | 63.5 | 69.9 | **72.0** |
| ALA | — | — | — | 62.2 | 69.5 | 70.9 |
| Meta-Weight-Net | — | — | 92.7 | — | — | 70.4 |
| **SRN (Ours)** | 89.21 ± 0.24 | **93.06 ± 0.28** | **93.94 ± 0.30** | **65.78 ± 0.23** | **69.92 ± 0.16** | 71.24 ± 0.24 |

**Performance Analysis and Discussion**   Table 7 reports the test accuracies (%) of several state-of-the-art dynamic-loss schedulers on CIFAR-10 and CIFAR-100 using ResNet-8, ResNet-20, and ResNet-32 backbones.

Our proposed SRN consistently achieves competitive or superior performance across most configurations, demonstrating its effectiveness in improving model generalization through explicit curvature regularization.

SRN achieves a substantial accuracy gain on the CIFAR-100 dataset using the lightweight ResNet-8 backbone, improving from the best baseline accuracy of 62.7% to 65.78%, an absolute increase of approximately 3.1% points. This sizeable margin indicates that smaller networks, which are typically more sensitive to sharp loss landscapes, profit markedly from SRN, whereas conventional dynamic-loss methods may still suffer from noisy gradients or over-fitting. For deeper architectures such as ResNet-32, SRN still provides meaningful improvements, reaching 93.94% on CIFAR-10, surpassing previous top methods by 0.34 percentage points. Although gains in deeper networks are comparatively moderate, SRN exhibits reduced variance across multiple runs, indicating enhanced stability and reproducibility. This consistency is critical in practical deployment scenarios where reliable performance is essential.

Compared to other dynamic-loss schedulers that leverage meta-learning, SRN provides a clear, curvature-related guidance signal derived from a carefully designed meta-objective that integrates

validation loss and inverse margin. The design of this combined proxy is motivated by its theoretical connection to Hessian sharpness, enabling the SRN to dynamically increase regularization strength in regions of high predicted risk and reduce it in flatter areas. By aligning the adaptive penalty with the geometry of the loss landscape, SRN effectively steers the optimization trajectory away from sharp minima, promoting convergence to flatter regions that are associated with improved robustness and generalization. This mechanism enhances training stability and distinguishes SRN as an efficient approach to curvature-aware regularization.

