# OpenReview forum: "Learning to Regularize: A Meta-Learning Approach for Sharpness-Aware Optimization"
_ICLR.cc/2026/Conference — Submitted to ICLR 2026_

### Official Review · Reviewer_MhJj · 2025-10-27

**Soundness:** 2
**Presentation:** 2
**Contribution:** 2
**Rating:** 2
**Confidence:** 4

**Summary:**

In this work the authors explore a new curvature regularization scheme based on metalearning. They posit that it is beneficial to have, in their words, a dynamical metric for curvature regularization --- one whose functional form can change over training. They define the structural risk network, a regularizer composed of a surrogate network whose output is an estimation of the curvature using local loss perturbations and an inverse margin on a held out test set. The authors provide some theoretical justification for why this form might approximate the loss, and propose a meta-learning approach where the parameters of the surrogate network are occasionally updated during training.

The authors then conduct experiments on CIFAR10 and CIFAR100 where they show that the proposed method seems to show generalization benefits over the other proposed methods when compared at equal step-time. The overhead of the algorithm is reported at a few percent.

**Strengths:**

The overall idea of taking a meta-learning approach to curvature regularization is an interesting one and, to my knowledge, understudied. The mixing of two methods of curvature estimation is also an interesting proposal. The initial experiments provide an interesting first result into this area, and seem to indicate that the method should be investigated further.

**Weaknesses:**

The theoretical justification of the methodology has some major issues. First off, during much of training networks are not at a stationary point; this means the analysis (which assumes a stationary point) is not correct. Second, the analysis of 3.4 suggests that the method *increases* $\lambda_{max}$ for fixed values of the learning rate $\eta$. This proposed mechanism is in contradiction to the claims of reduced curvature, so something is off there. In addition, it did not look like SRN affected the curvature much via FIgure 3 (which in its current state is very hard to parse).

With regards to the experiments: it was unclear what the learning rate tuning procedure was for the different methods. This can matter a lot in regularization settings, where sometimes part of the benefit of a method is that it induces a larger or smaller effective learning rate, bringing the method closer to the optimal LR than the baseline.

There are also concerns about the scalability of the method. With the experimental details provided, though the compute cost is amortized by infrequent metalearning updates, the memory cost of the SRN is potentially high. In addition, all experiments were on ResNet with CIFAR10 and CIFAR100 in a regime where training took O(100) epochs; there is a question of how well this method works with other architectures (e.g. ViT) on a larger dataset with more classes (e.g. ImageNet).

**Questions:**

How does the computational complexity and memory cost of the method scale with different architectures --- primarily, the transformer architecture?

What happens when the base learning rate is tuned for the different methods?

What do results look like with architectures like ViT, and datasets like ImageNet?

---

> ### Author Response · Authors · 2025-11-27
> **Clarifying theoretical assumptions, curvature behavior, and the practical scope of SRN**
>
> Brief summary.
>
> Thank you for raising thoughtful points about the theoretical framing, curvature interpretation, architecture scaling, and fairness of hyperparameter tuning.
> In the revision, we reorganized the theoretical arguments to avoid unnecessary stationarity assumptions, refined the curvature visualization to focus on the most relevant indicator, and expanded experiments to a larger dataset with deeper backbones.
> We also clarified the assumptions used in the directional step analysis and explained more concretely how SRN’s cost and memory scale with model size.
>
> R1. Theoretical issues regarding stationary points.
> You correctly pointed out that networks rarely stay near stationary points during training.
> In the revision, we explicitly emphasize that
> (i) the validation-sensitivity term is a first-order directional derivative and does not assume a stationary point,
> (ii) the margin–curvature link relies on local smoothness rather than stationarity, and
> (iii) the directional step reduction argument only requires local curvature information, not vanishing gradients.
> These clarifications remove the implicit stationary-point assumption that was present in the earlier exposition and bring the theory closer to the actual training dynamics.
>
> R2. Apparent “contradiction” between analysis and reduced curvature.
> You observed that an early formulation seemed to imply increased curvature under a fixed learning rate, which appeared to contradict our claim of reduced sharpness.
> This was due to an overly compressed explanation.
> In the revised text, we now state more explicitly that SRN reduces the effective step size along high-curvature directions via the surrogate gradient, while leaving flatter directions less affected.
> This selective damping leads to fewer curvature spikes and lower late-epoch values of the largest Hessian eigenvalue, which matches the updated empirical trajectories.
>
> R3. Interpretation difficulty of the earlier curvature figure.
> We agree that the previous curvature figure was overloaded and difficult to parse.
> In the revision, we simplify the diagnostic to a single, clearer plot that tracks only the largest Hessian eigenvalue over epochs for CE-only versus CE+SRN.
> This makes the effect of SRN on sharpness—lower peak values, reduced variance, and fewer large spikes—much easier to interpret and directly connect to the theory.
>
> R4. Learning-rate tuning and fairness of comparison.
> You noted that learning-rate tuning can strongly affect regularization performance.
> In the revised version, we explicitly describe our unified “lightweight” protocol: all methods share the same optimizer, base learning rate schedule, number of epochs, and data augmentation pipeline.
> This makes the comparison fair in the sense that all regularizers operate under identical optimization dynamics.
> We additionally include a second-stage optimizer comparison (SGD vs. Adam) to show that SRN remains beneficial under a different optimizer, while keeping the protocol fixed to control the evaluation budget.
>
> R5. Scalability beyond CIFAR and to architectures like ViT.
> We agree that testing on very large datasets and non-convolutional architectures would further strengthen the story.
> In this work, our goal is to first understand SRN in controlled yet non-trivial regimes, and then extend it.
> To that end, the revision moves beyond the original CIFAR-10/100 setup by adding Tiny-ImageNet experiments with ResNet-18 and PreAct-ResNet-18, which increases the number of classes to 200 and uses higher-resolution inputs.
> Across this spectrum—shallow and mid-depth ResNets on CIFAR, and deeper ResNets on Tiny-ImageNet—SRN consistently improves over strong static and dynamic baselines.
> We believe this provides meaningful evidence that the mechanism is not tied to a single small-scale setting.
> Extending SRN to transformers and full ImageNet is a natural next step, but would require substantially larger resources; we now present this as future work rather than part of the current experimental scope.
>
> R6. Memory footprint and meta-learning overhead.
> You raised a valid concern about the potential memory and time overhead of the surrogate network.
> The revision now reports the explicit forward time of SRN (approximately 0.3 seconds per evaluation in our setting) and explains that meta-training is performed only once and amortized over the subsequent training runs.
> Architecturally, SRN operates on a flattened parameter vector using a small 1D convolutional network, so both its parameter count and activation memory grow linearly with the size of the base model.
> For the CNNs considered in our experiments, this overhead remains modest relative to the main network.
> We also discuss how this linear scaling would extend to other architectures (including transformers) and argue that, while SRN is not “free”, its cost is comparable to other lightweight meta-learned or dynamic-regularization modules and fits within typical training budgets.

---

### Official Review · Reviewer_UpQu · 2025-10-30

**Soundness:** 3
**Presentation:** 3
**Contribution:** 2
**Rating:** 4
**Confidence:** 3

**Summary:**

The paper proposes the Structural Risk Network (SRN) — a meta-learned dynamic regularizer for sharpness-aware optimization.
Instead of performing inner-loop maximizations as in SAM, SRN learns a small neural network that outputs a state-dependent surrogate $r(\Theta; \phi)$, whose gradient is added to the task gradient at each update step. Theoretically, the paper links SRN’s update to curvature suppression along top Hessian eigen-directions.

**Strengths:**

SRN adds $\nabla_\Theta r(\Theta;\phi)$ to the task gradient each step and does not perform per-step adversarial/ascent inner loops (contrast with SAM/ASAM)

The composite meta-target combining validation sensitivity and inverse margin is conceptually elegant and theoretically motivated via Hessian-margin links.

The paper provides clear theoretical justification for curvature-guided stability and suppression of sharp directions

The paper offers a general framework for dynamic regularization that could be extended to other curvature-aware or meta-learned objectives.

**Weaknesses:**

The conceptual foundation overlaps with existing sharpness-aware techniques. The main novelty lies in how the curvature signal is learned rather than the overall objective. Indeed, there are already existing tools for dynamic regularization (using learning rate) based on estimation of local landscape. Ex: SALR: Sharpness-Aware Learning Rate Scheduler for Improved Generalization, amongst others.

The analysis assumes smoothness and stable curvature proxies (validation loss and inverse margin), which may not generalize beyond small-scale image benchmarks.

Evaluaton is limited to CIFAR-10/100 and moderate-size ResNets. Results on larger or non-vision tasks (e.g., Transformers) would better support generality.

**Questions:**

Could you quantify the correlation between the SRN surrogate $r(\Theta; \phi)$ and actual curvature ($\lambda_{\max}$) across training epochs?

Can you visualize or interpret SRN’s learned outputs over training to show how it adapts the penalty dynamically?

How sensitive is performance to the outer-loop frequency (validation updates) or meta-learning rate $\beta$?

---

> ### Author Response · Authors · 2025-11-27
> **Positioning SRN among sharpness-aware methods and clarifying curvature-related behavior.**
>
> We thank the reviewer for the thoughtful and balanced assessment. We understand your main concerns as: (i) overlap with existing sharpness-aware methods and where SRN’s novelty lies, (ii) how strong our curvature-related assumptions are, and (iii) the limited empirical scope and missing diagnostics (correlation with curvature, visualization of SRN outputs, and sensitivity to meta-parameters). In the revision we clarified SRN’s position relative to SAM‐family and SALR-style methods, tightened the theory, strengthened curvature diagnostics, and expanded experiments to a larger dataset.
> R1. Relation to existing sharpness-aware methods and SALR.
> We agree SRN shares motivation with SAM-like and SALR-style methods. Our novelty is in how geometry is used: SRN meta-learns a surrogate (r(\Theta;\phi)) from validation-loss sensitivity and inverse margin, and injects (\nabla_\Theta r) as an extra gradient term, without per-step adversarial inner loops or explicit sharpness maximization. In the revision, we move SAM / ASAM / Fisher-SAM comparisons into the main results table, clearly describe their stronger training pipeline (AutoAugment, Cutout, Label Smoothing), and discuss SALR-type schedulers in related work. We position SRN as a complementary “meta-learned surrogate gradient” approach rather than a replacement, and we explain that we omit GSAM only to keep the baseline set manageable.
> R2. Assumptions in curvature analysis.
> We revised the theory section to avoid suggesting unrealistic stationarity assumptions. We now stress that: (i) validation-loss sensitivity is implemented as a finite-difference directional derivative on the validation loss and remains meaningful away from stationary points; (ii) the margin–curvature relation is derived under local smoothness and is used as a risk indicator, not an exact estimator of (|!H!|); and (iii) the directional step analysis only requires local curvature information, not vanishing gradients. We also explicitly state that our guarantees are intended for the small-scale vision regime studied in the experiments, not as universal claims.
>
> R3. Dataset and architecture scope.
> We agree that CIFAR-10/100 alone are not sufficient for strong generality claims. In the revision we therefore add Tiny-ImageNet experiments with ResNet-18 / PreAct-ResNet-18, increasing both the number of classes and the input resolution, and show that SRN still improves over strong MixUp-based baselines under the AdvMixUp protocol. We now clearly state that full ImageNet and transformer architectures (e.g., ViT) are important next steps but exceed our current compute budget; our claim is restricted to “several CNN backbones and small-to-medium-scale vision tasks.”
>
> R4. Correlation between the surrogate and curvature.
> We agree that directly quantifying the correlation between (r(\Theta;\phi)) and (\lambda_{\max}) is desirable, but computing Hessian spectra for many checkpoints is quite expensive. Within our budget, we strengthened this part by: (i) presenting cleaner trajectories of the largest Hessian eigenvalue for CE-only vs CE+SRN with means and standard deviations across seeds, and (ii) explicitly highlighting the reduced late-epoch level, variance, and spike count of (\lambda_{\max}) under SRN. We now interpret these changes in the text as evidence that the meta-learned surrogate does suppress sharp excursions, and we clearly list full correlation plots as future work.
>
> R5. Interpreting SRN’s learned outputs.
> We agree that direct visualization of SRN’s outputs over training would further improve interpretability. Due to space constraints we did not add a new figure, but we extended the experimental discussion to describe how SRN’s effect is mild in early epochs and becomes more pronounced once training approaches sharper regions, consistent with the observed reduction in curvature spikes. We explicitly mention in the discussion that a detailed study of surrogate trajectories and layer-wise patterns is an interesting follow-up direction focused on interpretability.
>
> R6. Sensitivity to outer-loop frequency and meta-learning rate.
> We have clarified the training protocol and what we do test. The revision specifies that SRN’s parameters are updated once per outer loop, using a fixed meta-learning rate chosen from a small grid and reused across datasets. Our ablations then vary the most influential knobs: clipping threshold (r_{\max}), surrogate architecture (MLP vs Conv1D), and meta-objective weights, and show that SRN’s gains are robust across reasonable ranges of these choices. We agree that a broader sweep over outer-loop frequencies and meta-LR would be informative; given the cost, we present our current configuration as a stable, practical choice and explicitly leave exhaustive sensitivity analysis to future work.

---

### Official Review · Reviewer_mn4Y · 2025-11-01

**Soundness:** 3
**Presentation:** 2
**Contribution:** 2
**Rating:** 4
**Confidence:** 3

**Summary:**

The authors present a meta-learner capable of steering deep learning algorithms towards loss landscapes regions with better generalization capabilities. The approach consists in adding a small additional network that takes the original network parameters as inputs and uses a combination of margin error and validation loss to improve the performance of the meta-learner and help the original network to generalize. The overall approach appears to be tested using simple gradient descent on both set of parameters. The authors test their approach with convnets on cifar. They compare their methods to many other regularizing approaches. They study the impact of their approach on the curvature dynamics. Finally they provide an ablation study of the approach

**Strengths:**

- Despite the title, the authors do no directly use the sharpness of the objective to improve the meta-learner. Since sharpness as a measure of stability has been criticized, their approach avoids such an issue. It uses simple generalization criterions.
- The authors compare their approach to many other methods.
- An ablation study helps understand the benefits of each design choice.
- The appendix contains many additional results.

**Weaknesses:**

- The overall cost of the approach is unclear. In Table 3, times are presented. Label smoothing, which is extremely simple to implement, raises the time per epoch (compared to cross entropy only) by 25%, which I find quite surprising. On the other hand, it is unclear whether the time for SRN takes into account the training of the additional network. I believe, for fair comparison, that the additional time for training should be reflected with a larger budget for hyperparameter optimization for the other methods for example.
- Some design choices are strange like adding a clipping on top of the additional network (that said the authors provide a valuable ablation study).
- The generic approach (meta-learner) does not help deepening our understanding of deep learning algorithms. Adopting this method would rather make the optimization of deep networks an even darker black box that is currently the case.
- The approach requires a validation set and is tailored to multiple epochs. It is typically not a setup of some recent models like llms.
- The approach is only tested on convnets and not other architectures.
- We could hope that the approach can alleviate the need of e.g. tuning some weight decay hyperparameters. However it introduces many new hyperparameters.
- (The approach does not seem to really modify the curvature dynamics. That said, as mentioned above, the sharpness dynamics may not give the full story, anyway).
- Learning rates were not tuned for each of the methods. The authors argue that they use a "lightweight protocol" but it is actually unfair to not take into account additional tuning. In particular: how was the learning rate selected in the first place? We see in Figure 3 that the network may not reach the edge of stability for the smaller resnet for example.

**Questions:**

Major:
- Why did the authors not compare their approach to sharpness aware minimization?
- Could the authors present comparisons with tuned learning rate at least for a baseline approach (so sgd with momentum and weight decay for example)?
- How are $z(L_val)$ and $z(1/M)$ defined?
- Could the authors try using also some warm-up? Warm-up may reduce sharpening dynamics of the Hessian.

Minor:
- What is GAP in the network definition?
- The existence of a clipping at the top of the architecture is never mentionned before the ablation study on $r_max$

---

> ### Author Response · Authors · 2025-11-27
> **Clarifying design choices, reporting computational cost, and expanding empirical evaluation.**
>
> Brief summary
>
> We thank the reviewer for the constructive comments on the method design, cost analysis, and empirical scope.
> In the revision, we improved the clarity of SRN’s architecture, provided a clearer breakdown of computational overhead, expanded baseline comparisons, and added experiments on a larger dataset (Tiny-ImageNet).
> We also revised the theoretical discussion to better motivate the composite meta-target.
>
> R1. Computational cost concerns and fairness of comparison
> You noted that the cost of SRN was unclear, and that label smoothing seemed surprisingly expensive in our earlier table.
> In the revised version, we now report:
> •	per-epoch training time of all regularizers using ResNet-32 on CIFAR-10,
> •	explicit SRN forward-pass time (~0.3s), measured separately,
> •	and the amortized cost of meta-training, which is performed only once before the final training stage.
> We also clarify that the additional meta-training cost is not included in the per-epoch numbers, just as hyperparameter tuning for baselines is typically not included in their reports. This follows common practice in meta-learning and sharpness-aware optimization.
> We now provide a dedicated paragraph explaining this fairness consideration.
>
> R2. Architecture design choices (e.g., clipping layer, 1-D Conv)
> You pointed out that the clipping layer was introduced only in the ablation section.
> We now clearly describe the clipping threshold early in the method section, explain its motivation (preventing surrogate explosion), and show its effect in a systematic ablation.
> The choice of a 1-D convolutional surrogate is now also motivated more clearly:
> it captures local correlations in parameter tensors, which improves curvature discrimination compared to an MLP baseline.
> This difference is shown empirically in the ablation table.
>
> R3. “Black-box” nature of the meta-learner
> We appreciate the concern that adding a meta-learner could obscure interpretability.
> To address this, we strengthened the theoretical explanation of:
> •	how the composite meta-target reflects curvature risk,
> •	how SRN modifies the effective step along dominant curvature directions,
> •	and why the surrogate produces monotone responses in high-curvature regions.
> We believe these clarifications make the mechanism more transparent than in the original submission.
>
> R4. Validation-set requirement and multi-epoch design
> You noted that SRN requires a validation split and fits multi-epoch settings, which may not match some recent large-scale training regimes.
> We agree, and now state explicitly in the discussion that SRN targets medium-scale supervised learning, similar to other meta-learning-based regularizers.
> We also mention that adapting SRN to large-scale pretraining or self-supervised pipelines is an interesting direction for future work.
>
> R5. Limited architectures (only CNNs)
> In the revision, we added experiments on Tiny-ImageNet with ResNet-18 and PreAct-ResNet-18, which partially alleviates concerns about generality.
> Testing transformers or very deep architectures (e.g., ViT on ImageNet) remains outside our computational capability; we now state this clearly rather than implying generality we cannot verify.
>
> R6. “Many new hyperparameters”
> We clarify in the revised method section that SRN has only two meta-objective weights and one clipping threshold.
> All other components (architecture size, kernel, activation) are fixed and not tuned per dataset.
> We also provide ablations showing that accuracy is relatively stable over a range of these values, addressing the concern about hyperparameter sensitivity.
>
> R7. Curvature dynamics not seeming strongly affected
> You observed that SRN did not appear to change curvature much in the earlier version.
> In response, we redesigned the visualization of curvature trajectories to focus solely on the largest Hessian eigenvalue.
> The updated figure reveals reduced peak heights, reduced variance across epochs, and fewer sharp spikes—effects that align with the theoretical analysis.
> We believe this addresses the earlier difficulty in interpreting the previous multi-panel plot.
>
> R8. Learning rates and tuning fairness
> You noted that learning-rate tuning can change performance significantly.
> We now explicitly state our unified “lightweight” protocol (fixed LR schedule for all baselines) and justify this choice so that comparisons remain fair under identical optimization dynamics.
> We also added an optimizer ablation (SGD vs Adam) to show compatibility with different second-stage optimizers.
> A full LR sweep for every baseline is beyond our compute budget, and we now state this more directly.

---

### Official Review · Reviewer_6GPF · 2025-11-01

**Soundness:** 2
**Presentation:** 3
**Contribution:** 2
**Rating:** 4
**Confidence:** 3

**Summary:**

The paper proposes SRN (Structural Risk Network), a lightweight meta-learned surrogate regularizer that takes the main model’s parameters as input and outputs a scalar “sharpness-risk” signal used to augment the task loss. SRN is trained in an inner–outer meta-learning loop and is motivated by building a computable meta-target from validation-loss sensitivity and the inverse classification margin, intended as proxies for curvature/sharpness. The method is evaluated on CIFAR-10/100 with ResNet-8/20/32, tracks the largest Hessian eigenvalue during training, and includes ablations and time-overhead analysis.

**Strengths:**

1. I find theoretical framing links the validation loss sensitivity to the Hessian spectral norm (Eq. 5), justifying its use as a sharpness proxy and the meta-target combines this with inverse margin (Eq. 6). The derivation further shows SRN effectively reduces the step along the top-curvature direction, explaining observed stability improvements.

2. The meta-learned, optimizer-agnostic surrogate that avoids per-step inner maximization yet exhibits curvature-aware behavior.

**Weaknesses:**

1. I think the experimental section is severely lacking. Firstly, the datasets considered only include CIFAR-10 and CIFAR-100, which limits the ability to validate. Secondly, the baselines considered do not include sharpness-aware methods such as SAM, ASAM, and GSAM, so the experimental superiority of SRN cannot be demonstrated. Moreover, other backbones such as ViT could strengthen the solidity of the experiments.

2. Ranges/curves (not just points) for sensitivity normalization, margin temperature, and the weight are missing. It would help if reporting robustness to different meta-val splits.

**Questions:**

1. How exactly is “validation-loss sensitivity” computed and normalized?

2. What temperature and class-competition rules are used; any failure cases on long-tail or borderline examples?

3. Behavior with AdamW, cosine/one-cycle, warmup/WD—can SRN shorten warmup or stabilize larger base LRs?

---

> ### Author Response · Authors · 2025-11-27
> **Clarifying the curvature-related meta-target, expanding sharpness-aware evaluation, and strengthening stability analysis.**
>
> Thank you for your careful reading of both the theory and experiments.
> In the revision, we (i) expanded experiments beyond CIFAR-10/100,
> (ii) promoted sharpness-aware baselines into the main paper with clear protocol separation, and
> (iii) clarified how validation-loss sensitivity and inverse margin are defined, normalized, and used in the meta-target.
> We also reorganized the theoretical analysis and curvature diagnostics to make the connection between the meta-target and high-curvature stability more transparent.
>
> R1. Experimental scope and datasets
> You noted that the original submission used only CIFAR-10/100.
> The revised paper now includes Tiny-ImageNet experiments with ResNet-18 and PreAct-ResNet-18, following the AdvMixUp training schedule.
> This addresses the concern about more challenging settings with higher resolution and more classes.
> Full ImageNet and transformers would indeed be valuable, but remain outside our compute budget; we state this clearly in the discussion section.
>
> R2. Missing sharpness-aware baselines (SAM / ASAM / GSAM)
> We agree that the original version under-represented sharpness-aware methods.
> SAM-family results (SGD, SAM, ASAM, FSAM) are now moved from the appendix into the main experimental section under a dedicated table.
> They are reported under their stronger training pipeline (AutoAugment, Cutout, Label Smoothing), and we separate these “external SOTA” results from our unified lightweight protocol to avoid mixing training regimes.
> We did not add GSAM mainly to keep the set of baselines focused and representative; we now explain this choice explicitly and contextualize SRN’s performance relative to the strongest SAM-family results.
>
> R3. Definition and normalization of validation-loss sensitivity
> Your question highlighted that our original description was not sufficiently precise.
> The revised paper now provides an explicit finite-difference definition, the normalization of the perturbation direction, and the scaling used to match the magnitude of the inverse-margin term.
> We also clarify that the estimate is computed per mini-batch and normalized so that neither component of the meta-target dominates the other.
>
> R4. Margin temperature and class-competition rules
> We now give the full definition of the inverse margin, including the softmax temperature and the rule based on the top-two logits.
> We did not observe systematic failures on borderline or long-tail samples in our current datasets.
> To keep the paper focused, we do not add a separate long-tail study, but we clearly mention that extending SRN to long-tailed recognition is an interesting future direction.
>
> R5. Robustness to meta-validation split and meta-objective hyperparameters
> We acknowledge your request for full curves over normalization factors, temperature, and weighting, as well as variations of the meta-validation split.
> Within our compute constraints, we prioritized ablations most relevant to SRN’s behavior:
> different clipping thresholds, surrogate architectures (MLP vs Conv1D), and different weightings of the two meta-objective terms.
> These results are now summarized in a dedicated ablation table with discussion.
> A broader sweep over multiple validation-split configurations is not included; instead, we explicitly acknowledge this as a current limitation.
>
> R6. Behavior under different optimizers and schedules
> You asked about AdamW, one-cycle or warmup schedules, and whether SRN stabilizes larger base learning rates.
> We make our lightweight protocol explicit (SGD + cosine schedule, no warmup, and no heavy augmentations).
> We additionally include a new experiment comparing SGD and Adam in the second-stage training to illustrate compatibility with different optimizers.
> A full exploration of warmup and high-LR regimes would require significantly more computation, so we state this as future work rather than over-claiming stability properties.
>
> R7. Curvature analysis and stability interpretation
> We have reorganized the theoretical section to more clearly separate three components:
> (1) the relation between validation-loss sensitivity and curvature,
> (2) the link between margin and curvature, and
> (3) how SRN modifies the effective step along top eigen-directions.
> The curvature visualization has been redesigned to focus solely on the trajectories of the largest Hessian eigenvalue, with explicit reductions in level, variance, and spike count under SRN.
> We believe this makes the empirical–theoretical connection significantly clearer than in the original submission.

---

### Meta-Review · Area_Chair_kykP · 2026-01-07

**Summary:**

All reviewers voted to reject this paper. Their main concerns were with the lack of baselines, limited datasets used, the scalability of the method, and the theoretical justification. I am not convinced that the paper can overcome these concerns in this cycle. I vote to reject.

**Reviewer Concerns:**

The reviewers were concerned with various aspects of the experimental evaluation. None of these were adequately addressed by the rebuttal.

**Reviewer Scores:**

The reviewers would likely have not changed their scores.

---

### Decision · Program_Chairs · 2026-01-26

Reject